# Driving Cessation: What Are Family Members’ Experiences and What Do They Think about Driving Simulators?

**DOI:** 10.3390/geriatrics7060126

**Published:** 2022-11-11

**Authors:** Frank Knoefel, Salma Mayamuud, Rania Tfaily

**Affiliations:** 1Memory and Cognition Group, Bruyère Research Institute, Ottawa, ON K1N 5C8, Canada; 2Bruyère Memory Program, Bruyère Continuing Care, Ottawa, ON K1N 5C8, Canada; 3Faculty of Medicine, University of Ottawa, Ottawa ON K1H 8L6, Canada; 4Faculty of Engineering and Design, Carleton University, Ottawa, ON K1S 5B6, Canada; 5AGE-WELL National Innovation Hub—Sensors and Analytics for Monitoring Mobility and Memory (SAM3), Ottawa, ON K1N 5C8, Canada; 6Faculty of Science, University of Ottawa, Ottawa, ON K1N 9B4, Canada; 7Department of Sociology and Anthropology, Faculty of Arts and Social Sciences, Carleton University, Ottawa, ON K1S 5B6, Canada

**Keywords:** driving cessation, cognitive decline, driving simulators, qualitative data, caregivers, framework analysis

## Abstract

Background: Driving cessation is difficult for persons living with cognitive decline (PLWCD) and their caregivers (CG). Physicians are often required to notify authorities of driving risks, and typically base decisions on paper-based cognitive assessments and on-road tests. This study examines experiences surrounding cessation and CG’s views regarding simulators in the process. Methods: Semi-structured virtual interviews were conducted with CGs of PLWCD from an academic memory clinic. Experiences around cessation were explored first, followed by discussions regarding the simulator. Framework analysis was applied to transcribed interviews. Results: Six females and two males, three children and five spouses participated. PLWCD viewed driving cessation negatively, often had difficulty understanding why, and believed cessation was temporary. CGs experienced relief and/or shock. Cessation negatively impacted the relationships between the PLWCD and both the physician and CG. Isolation, coping challenges and loss of independence were experienced by the PLWCD. The lives of caregivers were adversely affected, especially regarding driving burden and worsening mental health. CGs were generally supportive of simulators. Positives included: measurement of driving skills, method of testing, and providing an understanding regarding the driving suspension. Potential drawbacks included difficulty using the machine, testing anxiety and stress induced by a crash. Caregivers were concerned about: PLWCD’s disappointment of failure, requesting to retest, and reluctance to accept the decision. Conclusion: PLWCD and caregivers had negative experiences related to the driving cessation. Generally, caregivers viewed implementing driving simulators positively, in a context of a practice session and support for PLWCD’s potential reactions to the decision.

## 1. Introduction

It is a challenge to balance supporting older drivers to maintain their independence and, hence, wellbeing with the rights of other users of the road, including drivers, passengers, and pedestrians, as well as facilities around roads to be safe from harm caused by an automobile crash. Decisions trying to balance these competing interests play out everyday in physicians’ offices around the world. In many jurisdictions physicians are required to notify driving authorities about driving risk [1]. 

There is extensive literature showing the impacts of aging on cognitive domains important to driving safety [2]. Similarly, there has been a lot of work looking at the impact of cognitive decline on driving safety [3,4]. At the same time, a number of studies have documented the negative impact of driving retirement on older adults’ wellbeing. These can include declines in general health and physical, social and cognitive functioning, possibly leading to institutionalization and even death [5]. 

The qualitative literature in this area has shown that the actual events around the notification of driving cessation can be quite traumatic to the person being reported to a driving authority and their family. Adler and colleagues suggest that few drivers with cognitive decline and their families prepare for driving cessation [6,7]. There is often a sense of reluctance and loss associated with driving cessation as well [8,9]. In fact, some drivers with cognitive impairment react with shock and anger when discussing their driving risk [10]. This has led to one group to suggest that “Advance Driving Directives” would be a good way to prepare for this eventuality [11] (p. 1573). Consequently, more work needs to be done, especially to translate this research into practical guidelines for frontline physicians, to improve driving risk assessments and to evaluate the potential of driver rehabilitation. 

In addition to the clinical assessment, including paper-based cognitive tests, performed by physicians, there are three traditional ways to evaluate driver safety: on-road testing, simulator testing and naturalistic driving [12]. Each has its advantages and disadvantages. On-road testing has the advantage of decades of experience and being the gold standard, but has the disadvantages of differences between examiners, road routes, test vehicles, weather, and actual events during the drive. Naturalistic driving has the advantage of monitoring drivers in their own vehicle and natural environment but has the disadvantage of being limited by the drivers’ routines during the assessment period [13]. For instance, if during the recording period the driver avoids rush hour, highways and inclement weather driving, there is no data available to assess driving abilities in these settings. Driving simulators have the advantage of being able to standardize routes, the environment and driving events, but may not feel like real driving, need more validation, and can cause simulator sickness [14]. Driving simulators have been used for driver training and rehabilitation, but this requires more validation work as well. 

While driving simulators have value in driving assessment and training/rehabilitation, there is a gap in the literature and not much is known about what older drivers with cognitive impairment and their family members think about this method of assessment. In this study, it was decided to interview family members of persons living with cognitive impairment that had recently had an appointment with a memory clinic physician where the outcome was that a letter would be sent to the driving authority. This group would have recently experienced the challenges of driving cessation and would have an important perspective on the addition of a driving simulator to the appointment process. Therefore, the objectives were to (1) get further understanding of the impact of the driving cessation discussion and (2) to determine the opinion of the family members regarding the inclusion of a driving simulator in the driving assessment of their relative with cognitive decline. 

## 2. Data Collection and Analysis

We used a qualitative research approach to better understand the experiences of caregivers regarding driving assessment and cessation of the persons with cognitive decline and to gauge the caregivers’ reactions regarding the use of driving simulators to assess driving. 

### 2.1. Study Setting

The Bruyère Memory Program is the only academic medical clinic devoted exclusively to cognitive assessment of older adults in Ottawa, Ontario, Canada, a city of about 1 million inhabitants. Patients are referred from family physicians, as well as other specialists. They are assessed by a clinic nurse and then a physician and may be referred to a neuropsychologist for more in-depth cognitive assessment. There are three types of physicians at the clinic: cognitive neurologists, geriatricians, and family physicians with additional training in care of the elderly. Driving risk assessment may be included in the reasons for referral but is automatically done for each patient referred to the clinic. 

### 2.2. Participants

Family members of aging adults with cognitive decline who had recently been reported to the provincial driving authority (Ministry of Transportation of Ontario) were recruited to participate in this study as they have a unique perspective regarding driving assessment possibilities. The protocol thus involved approaching the family member who had attended the appointment when driving risk was discussed between 6 to 24 months before the time of the study. This window was chosen so that the family members would be out of the ‘immediate’ impact time, but not be too long to remember the details. Generic consent to contact for research purposes is usually part of the admission process for the clinic. Two physicians at the clinic keep detailed copies of the letters sent to the driving authority. 

The steps to identify potential participants were as follows. The first step was to review each letter sent by the physicians. The next step was to determine if generic consent to participate in future research studies had been obtained. Those who had provided generic consent were contacted by the research assistant and asked for specific consent to participate in this project. Participants were told that participation is voluntary, and their involvement would be about a one-hour interview using video-conferencing technology. Specific consent was sought for the auditory recording of the interview. 

### 2.3. Data Collection 

Semi-structured interviews were carried out via the Zoom teleconferencing technology (in light of the COVID-19 pandemic and public health measures). An interview guide was developed to provide structure to the interviews and ensure that all relevant topics are covered. The interviews were divided into 2 sections. First, the participants were given the opportunity to talk about their experience with the appointment when driving was discussed and the weeks that followed. They were cued to recall the reactions of the person with cognitive impairment during that period and how they coped, and then to talk about their own experiences as a caregiver during that same period. 

Before continuing, the participant was then shown a short video clip that showed a Virage VM500 Car Driving simulator, and how it can be used [15]. The VM500 uses advanced simulation software and a driver environment built with actual car components providing a realistic feel of all controls. It includes 3D sound and high-fidelity motion. This video was selected as it represents the driving simulator that will shortly be installed at the Elisabeth-Bruyère Hospital in Ottawa, Canada.

The second section then focused on the participant’s perceptions of a driving simulator and how the addition of a simulator to the driving assessment might impact the person living with cognitive impairment and themselves. To keep the conversation focused, they were cued about potential advantages and disadvantages of coming to an additional appointment, the use of technology specifically, and potential reactions of the person living with cognitive impairment. See Appendix A for the interview questions.

The duration of the interviews varied between one hour and one and a half hours. The audio-recorded interviews were transcribed verbatim.

### 2.4. Data Analysis

Data were analyzed using framework analysis which offers a structured way to manage data, identify themes and develop explanations [16]. We followed the steps of framework analysis: familiarization, identifying a framework, indexing, charting; and mapping and interpretation [16,17,18]. Coding was done using an inductive approach, based on the content of the transcripts. The process of developing codes and themes was iterative, with the coding and recurrent themes discussed on a regular basis by the research team. NVivo was used to assist in data management, tagging data extracts into codes and organizing codes and themes. Excel was also used to summarize relevant data per participant to compare emergent themes across and within cases.

To improve rigour in our study, we adhered to several practices outlined by Green and Thorogood [19]. We conducted interviews with caregivers with intimate experiences regarding PLWCD’s driving cessation. Each of the authors read each of the interview transcripts numerous times to familiarize ourselves with the whole data set. Coding and identifying emergent themes were done separately by each of the authors and then discussed as a group. As discussed previously, we followed the procedures of framework analysis which provides a rich description regarding data management and identifying themes, and we kept an ’audit trail’ in Word and NVivo. In describing our findings, we provided ‘thick description’ in the form of numerous quotes from various participants and context to ground our interpretations, and we accounted for cases which deviate from the dominant theme(s). As outlined by framework analysis, we compared our themes across cases and within cases as well as compared our findings with those reported by other studies. 

## 3. Results

The original intent of the interviews was to determine family members’ opinions on the use of a driving simulator as part of a driving assessment. It was felt that it would be important to begin with a chance for the family members to remember the appointment that led to the driving cessation letter, and to use that context to think about the possibilities of a driving simulator assessment. This reflection on driving cessation ended up taking up over half of the interview times. 

### 3.1. The Participants

During the period June 2019–December 2021 there were 80 letters sent to the Ministry of Transportation of Ontario by 2 physicians at the Bruyère Memory Program. After review, 5 were removed because of particularly difficult lived experiences. Of the remaining 75, 32 had agreed to be contacted for future research. So, the number of potential participants in this study was 32, and we contacted all of them. Of those, 9 agreed to participate in the interview, and 8 (or 25%) were successfully scheduled for interview between April and July 2022. The other potential participants either informed us that they could not participate because of their busy schedules or did not contact us back. 

Based on the eight participants who were successfully interviewed for this study, there were four male and four female older adult drivers with cognitive decline. Seven were reported to the Ministry of Transportation, and subsequently stopped driving, and one did an on-road test that they passed. 

The interviewees consisted of 6 females and 2 males with an average age of 62.1 years (55–73). They identified themselves as 5 spouses and 3 children. Four identified as having a vocational college education and 4 had gone to university (Bachelors and Masters). They self-described as having been closely involved in the care of their loved one for an average of 3.4 years (2–9). Table 1 lists some of the demographic information of participants and the relationship to the PLWCD. The driving status of the PLWCD is not included to protect the anonymity of the participants. It is worth noting that we did not notice any differences in terms of caregivers’ reports about their own reactions and the reactions of the PLWCD to discussions about driving cessation between the PLWCD who eventually passed the road-test and those who stopped driving.

The participants were asked about their experiences with the period before and after the appointment when they were informed that a letter would be sent to the driving authority. They described their reactions and the reactions of the person living with cognitive decline as well as the impact on their lives. See Table 2. 

### 3.2. Reactions of Persons Living with Cognitive Decline to Driving Suspension

Persons living with cognitive decline (PLWCD) generally experience driving suspension as a very significant development with profound implications. Three sub-themes were identified in terms of the reactions of PLWCD based on interviews with caregivers: negative emotions, difficulty in comprehending the suspension and thinking that the suspension is temporary.

#### 3.2.1. Negative Emotions

Each of the caregiver participants have noted that their family member living with cognitive decline reacted negatively to the driving suspension. The extent of the negative emotions differed from one PLWCD to another; however, each one was unhappy and frustrated that they could no longer drive. The reactions ranged from being upset and frustrated to “extreme anger”. One of the participants referred to the driving suspension as traumatic to the PLWCD; another as deeply hurtful to the PLWCD: 

“*He fished, he had a boat, he had a truck, and he could no longer do that and that was a love of his. So, he was extremely upset, and I don’t blame him. Yeah, it was not a good day. During the appointment, he wanted to leave because he was very mad*.”(P3)

“*For him, it was like you took his life away because he was very independent before so that was just awful how it came down … He was very angry in the session like extremely angry*.”(P68)

The negative reactions tended to stronger in the cases of men living with cognitive decline, with three of them (but not the fourth) expressing extreme anger, verbal abuse or banging on tables. Some of the women participants invoked gender to explain the reactions of their husbands: 

“*I think for a man it means so much more. That really hurt him*.”(P68)

#### 3.2.2. Difficulty in Comprehending the Suspension

The negative reactions of persons living with cognitive decline were often compounded by the fact that they generally did not understand the rationale for the driving suspension and/or the connection between the written assessment (memory tests) and driving. None of the PLWCD in this study agreed with the decision that they are no longer capable of driving; they generally held on to the belief that they could, especially that they have been driving for years, and that they should not lose their license: 

“*Well, she was frustrated and quite upset because she didn’t see any reason for it. She did not see the risk that she was taking*.”(P15)

“*It had a really strong impact on her about losing her license and not understanding how the tests related to now you can no longer drive from the progression*.”(P87)

#### 3.2.3. Thinking It Is Temporary

Possibly related to not understanding why their driving privileged are taken away, many held on to the belief that the suspension is temporary, and that they will get their licenses back. Some actively inquired about the steps to get it back; however, none of the seven persons living with cognitive decline who lost their driving license followed through. There were a variety of reasons for this, including discouragement from caregivers and the cost of doing the road test:

“*…But her second reaction to that was how do I get it back. What do I need to do son to make sure I get it back*?”(P54)

“*She really was concentrated on how to get it back … As soon as she heard it was gonna cost money to do, she didn’t fight as hard to take the test…*”(P49)

### 3.3. Caregivers’ Reactions to Driving Suspension

In terms of caregivers, the most common reaction was a feeling of significant relief that the PLWCD would not be driving anymore. A minority of caregiver participants (all three were women caring for their husbands) experienced shock and stress as they were not prepared for this and had to deal with their husbands’ extreme anger. 

#### 3.3.1. Relief

Many of the caregiver participants expressed feeling relieved as they believed that it is not safe for the PLWCD to drive anymore. They were concerned about potential accidents in case the PLWCD continued to drive (e.g., being hurt or hurting someone), or being lost during driving. Many of the caregivers believed that it was the right decision, and that it allowed them to sleep better at night even though they were aware that they would have to do more in terms of driving the PLWCD to appointments, shopping, etc.:

“*It was a relief. My mom had been in a very risky situation … Her physical condition was such that driving was not safe. … it was a relief to hear that she would be off the road, even if it meant that I was gonna be doing her driving.*”(P15)

“*That was a relief to me that she wouldn’t be out there driving, either causing an accident, being in an accident or maybe even getting lost …*”(P87)

This reaction of relief is not surprising given that all eight caregiver participants stated that they had concerns regarding PLWCD’s driving or being lost while driving prior to the doctors’ assessments. Two of the caregiver participants had themselves raised concerns about their mothers’ driving with the doctors, another stated that she could see herself being the one raising concerns about her mom’s driving to the doctor, a PLWCD herself informed the doctor that she was getting lost while driving which got the ball rolling leading to the suspension of her driver’s license (the PLWCD later expressed regret for raising this issue with her doctor). In terms of the men living with cognitive decline, in three cases, the wives believed that the husbands’ driving prior to suspension were bad or scary or not safe, and the fourth wife had concerns about her husband getting lost while driving: 

“*It was a relief (laughing). The last time I drove with him was really scary, so it was a relief.*”(P22)

“*I had called the GP to say that I was very concerned with her driving …*”(P15)

#### 3.3.2. Shock and Stress

While most caregivers in this study had expressed relief over the PLWCD’s driving suspension, and all caregivers pointed out that they had misgivings related to the PLWCD’s driving prior to the suspension, three of the caregivers stated that they experienced shock and stress when the PLWCD were informed that they could not drive anymore. Several factors account for this finding. The caregivers were sometimes not aware that the appointment the PLWCD had at the memory clinic would include assessments that would put his/her driving license at jeopardy; hence, feeling unprepared. Caregivers would also agonize about the future, given the uncertainties, issues regarding transportation, and what life would look like with such life-altering suspension and diagnosis. A contributing factor in the above three cases (all women) was their husbands’ harsh reactions and outbursts during the appointment and afterward. The wives had to deal with extremely angry husbands which exacerbated their feelings of shock and stress:

“*It was a shock for me … We weren’t prepared … I was shocked by that decision. Right then and there I didn’t see it coming… and then I had to deal with his feelings. I couldn’t see what our future was gonna be I was just trying to take care of him and then he was so angry …, And I couldn’t see the next week in the calendar I was just living in that moment and the moment was awful.*”(P68)

“*It was actually very stressful because we moved … to the country. It’s a drive to come out to our place now. Literally, we bought this house a week before he was diagnosed with mild cognitive impairment … we need transportation. There’s no public transportation out here. There’s nothing. At the same time, I was having some eye issues and thinking OMG if he loses his license I am not driving because I can’t see very well, and we were up a creek (laughing) so yeah it was very concerning to me.*”(P73)

### 3.4. Impact of Driving Suspension on PLWCDs

Persons living with cognitive decline generally experienced driving suspension as a shock and an important disruption to their lives, activities, and routines. It was a turning point, signifying that life as they knew it was coming to an end. Driving suspension had significant impact on the lives of the PLWCD in terms of loss of independence, loss of control, isolation, and coping struggles. 

#### 3.4.1. Loss of Independence and Loss of Control 

Persons living with cognitive decline generally equated driving suspension with loss of independence. Not only have they lost the freedom to drive themselves wherever and whenever they wish, but they also have had to depend on others to drive them to appointments, shopping, activities. They often need others to find the time for them within their busy schedules. Such dependency did not sit well with many of them. In addition, driving suspension robbed some from the opportunity to take part in activities that they enjoy. A number of caregiver participants linked PLWCDs’ negative reactions to the suspension to their loss of independence: 

“*I mean you lose your license for something like this, and you’re sick you’re not comprehending it, you get angry, and you also lose your freedom a lot because now you’re relying on other people to get you here and there.*”(P3)

“*I know she was angry, she was frustrated, she realized that life as she knew it was no longer going to be what she knew. It was going to change, and it has, and it did*.”(P87)

Driving suspension is also associated with loss of control. This is most obvious in the case of vehicle(s) that the PLWCDs previously had control over. In some cases, the caregivers sold the vehicles or moved them to other locations, lest the PLWCDs forget about the suspension and drive which did not sit well with the PLWCDs:

“*He was so upset, and there sat the truck and the boat in the driveway, and I had to sell them because it didn’t make sense not to. He was really upset about that. A couple of months after the boat and truck were gone, he still gets upset … “He’ll say things like oh yeah, I used to have a boat and a truck, but it was taken away from me …*”(P3)

There was a notable exception to this. A PLWCD who would insist on keeping the car keys in his possession and would only relinquish them temporarily. This is not surprising considering that his wife believed that he did not fully understand that he will never regain his driver’s license: 

“*…But I would take his car, go to the garage, have the maintenance done, come back and as soon as I would get inside the door, he would ask me for his car keys …*”(P22)

#### 3.4.2. Isolation

Driving suspension could also lead to declines in socialization and feelings of isolation. Some of the PLWCDs perceived that they were isolated simply because they are not allowed to drive: 

“*It wasn’t good at all … He couldn’t drive so he wasn’t going anywhere. He was very angry with me still.*”(P68)

“*She thought, she probably still does, that she was stuck in her apartment because nobody would let her have her car and as soon as she had her car, she would be free to drive anywhere.*”(P49)

Isolation is also exacerbated by cognitive decline as friends and family members distance themselves: 

“*… people, friends don’t know how to deal with someone with dementia. Even though they would come and visit her before, some of them now don’t even visit. Don’t even call …*”(P87)

It is worth noting that isolation due to suspension of driving was not an issue in two cases in which the PLWCD was not that dependent on the car. This is the case for PLWCD who moved to assisted living which enabled her to make use of the residence’s transportation and the case of the PLWCD who lives in downtown, with easy access to various amenities:

“*She was a very independent person up until that point and she also has a certain personality type where she doesn’t like asking for help. I think her way of coping was to agree to completely change the situation so that she was in a residence and wouldn’t need to be doing things like groceries or she would have transportation available at the residence … so that she could go independently to appointments ….*”(P15)

“*Fortunately for her, she lives in downtown Ottawa. She could walk to everything, so we would take her we would walk her to get groceries, or we would walk her to go to church at noon, so she wasn’t in a situation where she was so stuck that she couldn’t go anywhere that she had to have her …*”(P54)

#### 3.4.3. Coping Struggles

Many of the PLWCD struggled to cope with the loss of independence and with feeling isolated. This group of PLWCDs was also particularly vulnerable because the driving suspension for some of them had occurred during the COVID-19 pandemic when visiting others or receiving visitors were restricted: 

“*It took about 2 years for it to settle in and accept the fact that I have to drive him wherever we go. He doesn’t go out a lot.*”(P3)

“*I think the trouble is she didn’t cope better, and it doesn’t help that we were under lockdown for so long so not only could she not drive anywhere … but she didn’t have daily visits from anybody … Everybody was keeping to themselves, the difference being the friends could play on the internet, go on google, watch Netflix, and my mother couldn’t do any of that … The timing of everything couldn’t have been worse.*”(P49)

With the passage of time, the reactions of the PLWCD tended to become milder, with anger subsiding. This was either because the PLWCD became more at peace with the decision and/or because the cognitive decline was robbing the PLWCD of their memories regarding the loss of their driving license. Two of the caregivers even reported how after considerable time has passed, the PLWCD would start to make fun of the fact that they could no longer drive: 

“*The funny thing is he still has the notion that he’s kind of driving, so he’s always going for the driver’s side to get in the car (laughing), but then he says oops no. So, he turns around and he goes to the passenger’s side.*”(P22)

“*Oh, this is funny … Nine times out of 10 if I’m walking towards the passenger door to open it, she looks at me and says you want me to drive, with a big smile on her face (laughing). So, we have the standing joke every time now. I guess telling you that she’s probably way more accepting of it now, but honestly that’s almost 2 years later.*”(P49)

### 3.5. Impact of Driving Suspension on PLWCD-Doctor Relationship

Even though the caregivers were not asked specifically about the PLWCD-doctor relationship, most of the caregivers reported that the PLWCD was angry with the doctor who had told them that he/she would report the PLWCD to the Ministry of Transportation (MOT), and all four men living with cognitive decline (but none of the women) had refused to talk to or see the doctor again. The driving suspension seems to have destabilized the PLWCD-doctor relationship. It is also possible that the PLWCDs viewed being reported to the MOT as a form of betrayal.

#### 3.5.1. Anger at the Doctor

Most of the PLWCDs held the doctor responsible for the suspension of their driving privileges, and hence, were angry with him/her. This is particularly because driving suspension has profound impact on the PLWCDs:

“*Well, I think running through all of this, throughout was the fact that she was angry. She was angry with [the doctor] …*”(P87)

#### 3.5.2. Refusal to See the Doctor Again

Each of the four men living with cognitive decline in the study refused to go see or talk to the doctor, including the man who did not end up losing his license as he passed the road test. However, we do not have such reports regarding any of the four women living with cognitive decline even though two of the women were reported to have been angry at the doctor (in one case, it was only after the doctor took the blame as the woman was blaming her daughter). However, even these two women who were angry at the doctor seem to have gone back and seen the doctor. It is possible that driving means much more to men, and it is perceived as a marker of masculinity and manhood rather than just a mean that enables independence in terms of going places. One of the caregivers (P68) stated that her husband’s loss of his driving license was akin to having his life taken away from him, and that it “one of the worst things [one] could do to him”: 

“*… because after that we still had to go see the [doctor] for other stuff because he didn’t have a diagnosis yet and he didn’t want to go to the [doctor] because he was mad.*”(P22)

“*He won’t even now go and see [the doctor at the hospital] because he blames him. He’s mad at him and he’s mad at me for losing his license.*”(P3)

“*And I can’t get him back to the [hospital], he won’t go back and not to see anyone. And [the doctor] kindly offered to give his file to another doctor you know. He wants nothing to do with [the hospital] unfortunately.*”(P68)

“*My husband didn’t want to talk to him [the doctor]. He was so mad at him, and he didn’t want to see him or talk to him.*”(P73)

### 3.6. Impact of Driving Suspension on PLWCD-Caregiver Relationship

Driving suspension also impacts the PLWCD-caregiver relationship, with caregivers bearing the brunt. PLWCDs often blamed the caregivers for losing their driving licenses and were angry at them. Driving suspension also leads to considerable conflict between the PLWCD and the caregiver which lead to increased stress in the relationship.

#### 3.6.1. Blame and Anger

Most of the PLWCDs blamed the caregivers and/or were angry with the caregivers regarding driving suspension. The caregivers were blamed for raising concerns with the doctors and/or answering the doctors’ questions during PLWCDs’ appointments. The PLWCDs were often angry at the caregivers because they would remind the PLWCD that they are not allowed to drive and would be in charge of making sure that the PLWCD does not drive. In addition, PLWCDs’ loss of independence and control and increased dependency on the caregivers likely exacerbate feelings of resentment and anger. The anger directed at the caregivers was sometimes extreme, especially in the case of three women caregivers who were caring for their husbands:

“*On the way home, in the car, he started cursing at me (laughs). He would never hurt me. He’s never physical, but verbal, yes. It was a long car ride home (laughs). It was not good. It was bad.*”(P3)

“*I’m trying to encourage him to get out of his real state of anger. I mean really because it was directed towards me … I’m all alone here caring for him because [the] kids live away.*”(P68)

“*It was an awful day. It was terrible. We had our appointment with [the doctor] and when we left there, he would not speak to me. He was just livid. He didn’t speak to me probably for 3 days that week … He was so angry and every time he’d go by in the house he’d bang the table, or he’d bang the counter. It was terrible ….*”(P73)

“*… she was angry with me when I would say I have to drive you. I’m just gonna say she was angry. There’s no other way of putting it. It’s much stronger than a disappointment. She was angry that she could no longer drive.*”(P87)

#### 3.6.2. Tension and Stress

Driving suspension also often led to increased conflict, tension and stress in the PLWCD-caregiver relationship. The caregiver has to deal with PLWCD’s negative feelings about suspension, manage PLWCD’s anger, pick up the pieces, and chart a new path forward. This has considerable impact on the caregivers, especially that many of them lack strong support networks, and they have to manage mostly on their own:

“*Well, the short term was pretty hellish. Mainly, because in that period of time there was conflict between my mom and I that resulted in the loss of her license. It increased anxiety and tension between the two of us.*”(P15)

“*We’ve been to hell and back with him losing license. I wouldn’t do anything to remind him…*”(P3)

“*He was livid. We got home and said it was all my fault, that I threw him under the bus, and I never should’ve answered questions … He was just awful. It was awful … Let’s say for 36 h it was pure awful.*”(P68)

“*It was a lot of stress. It was very stressful and very tense in the house. It was not nice. It was not good.*”(P73)

Despite these negative experiences, some of the caregivers expressed compassion toward the PLWCD as it is not the PLWCD’s fault. Only two participants did not mention any negative impact on the PLWCD-caregiver relationship in terms of blame or anger directed at the caregiver or increased tension and stress. In both cases, the PLWCD (one man and one woman) had expressed relatively mild negative reactions regarding the suspension.

### 3.7. Impact of Driving Suspension on Caregivers

In addition to its negative impact on the PLWCD-caregiver relationship, PLWCD’s driving suspension adds to the caregiver burden and worsens caregivers’ well-being. This is because driving suspension often results in the caregivers spending substantial amount of time driving PLWCD on top of the caregiving that they were doing. This exacerbates feeling of exhaustion and mental health struggles, places restrictions on one’s social life and increases conflict between work and caregiving.

#### 3.7.1. Driving Burden and Exhaustion

All women caregivers of PLWCDs who lost their driving licenses (with one exception) reported strong negative feelings regarding the driving they had to do, noting that driving takes time; that they are often on the road, or that they resent having to drive so much:

“*I’m on the road all the time. I hate driving now. It’s impacted me greatly because I am on the road. My –[child] doesn’t drive so I have to take them to all their appointments.*”(P3)

“*I resent having to be the one to drive her everywhere. Ironically, growing up she said to me as soon as you turn 16 and get your license you will be driving me around because I drove you for 16 years. Anyhow, I’m well passed paying her back for 16 years of driving, well passed that. Every now and then I’ll remind her don’t you like being chauffeured around.*”(P49)

“*Back in November, I didn’t know what was coming my way at all other than I would have to drive. I hate driving. I can’t believe I gotta drive, and I gotta drive with him who’s so angry and criticizing everything.*”(P68)

The driving burden along with other caregiving duties exacerbated caregivers’ feelings of exhaustion. Some of the women caregivers longed for a break because of how tired and exhausted they felt, with some experiencing additional pressure due to competing needs such as having to care for other family members. Feeling of exhaustion is particularly acute in the case of caregivers who do not get much help.

#### 3.7.2. Life-Caregiving Imbalance

The driving burden also had a significant impact on some of the caregivers’ social life and/or work. Some of the women caregivers lamented that their caregiving duties prevented them from visiting other family members or from having a social life or paid work. Given the impact of driving burden and other duties on caregivers, it is not surprising that some reported struggling in terms of their mental health:

“*I’m very close to my sisters and brother, and I wanna go home and I’m angry that I’m not sure whether I’ll be able to …. I finally get retired, and I don’t have the freedom to do simple things … Everybody needs a break no matter how much you love somebody ….*”(P3)

“*, there were numerous appointments that she had to go through, so I was juggling this while working full time and having to take holiday time to be running my mom around. … Socially, when you’re done running around the place, you’re exhausted so my social life definitely took a hit. I also have a mental health condition myself and it destabilized me … By Christmas, I would say I was… burnt out.*”(P15)

Trying to maintain a balance between personal life and caregiving duties can come with costs, for instance in the form of guilt:

“*I’ve felt increased guilt. The only social impact it might have is I definitely see her every single Saturday because she has a weekly hair appointment. I pick her up, we go to the hair appointment, we go to lunch after and spend a few hours in the afternoon. By the time I need to go home, I also need that glass of wine … I’d have to say the number one emotion is guilt over not seeing her as often as I should. I’m an only child so that doesn’t help …. I still see my friends, but I feel guilty while doing it.*”(P49)

It is important to note that one of the women caregivers did not report any negative impact due to her husband’s driving suspension as she and her husband had already been going everywhere together. The only change was that she has been the driver rather than him. Additionally, neither of two men caregivers identified driving suspension as the factor exacerbating exhaustion or negatively impacting their work or social life:

“*She has access to 3 different adult day programs. I gladly take her, gladly drive her, pick her up, bring her back home. Even doctor’s appointments, lab tests so to me that has not had any impact on me at all. It is what I would do no matter what. Her losing her license has not had an adverse effect on me. Her dementia, however, has.*”(P87)

### 3.8. Using the Driving Simulator to Test Driving Skills

As described in the Methods, after reflecting on their past experiences, the participants watched a video of a driving simulator. The second part of the interview resulted in the following themes: usefulness of the driving simulator in retrospect, driving simulator as beneficial, drawbacks to the use of driving simulators, and perceived reactions of the PLWCDs to the use of the driving simulator. See Table 3.

The caregivers were generally supportive of the use of a driving simulator. However, there was variation in terms of their support. Some of the participants voiced unqualified support for using the driving simulators, others provided conditional support or stated that while the driving simulator is a good idea, it is not for everyone.

#### 3.8.1. Unqualified Support

A number of the participants were enthusiastic about the use of the driving simulator in assessing the driving skills of persons living with cognitive decline and believed that it would have been a great option when their family member with cognitive decline was undergoing tests for driving assessments:

“*… personally, I think it’s a brilliant idea. It probably would’ve saved us a lot of heartache and stress.*” (P15)

“*But the simulator would’ve been nice because it’s the parameters are more set; it’s more, it’s computerized … It’s definitely a yes.*”(P22)

#### 3.8.2. Conditional Support

Some of the other participants were, generally, in support of using the driving simulator to assess driving skills, but they believed that other conditions should also be met for transparency and fairness. Participant 68 believed that the driving simulator offered “many benefits” given the increase in the number of people living with cognitive decline. She indicated that if her husband had to choose between being told to stop driving based on the cognitive test and doing the simulator test, he would opt for the latter. Nonetheless, participant 68 was adamant that there should be prior disclosure to the caregiver and aging adult about the driving simulator and the implications of failing the test because caregivers would be dealing with any potential fallout. As discussed previously, participant 68 felt blindsided with her the driving suspension of her husband who reacted strongly and angrily. It is worth noting that participant 68 believed that if her husband were to know that his driving would be assessed during the doctor’s appointment, whether through written test or driving simulator, and could be at jeopardy of losing his driving license, he would not have come to the doctor’s appointment.

“*… There would have to be advanced disclosure to the client and the caregiver. So, to know that this would be assessed, and this could be a factor. I’m not sure if that would reduce the number of people going to the memory clinic or not. In my case, I would be wholeheartedly supportive … Whatever the decision is and if it’s not in favor of the patient, it’s the caregiver that has to pick up the pieces and depending on the relationship, it could be a very difficult path.*”(P68)

Participant 68 also raised the issue of allowing PLWCD to get comfortable with using the simulator before the test especially that the concerned group is aged 65 years and older. Similarly, participant 73 was supportive as long as the person living with cognitive decline is given an opportunity to familiarize himself/herself with the simulator prior to the test. While she called the use of the driving simulator “a fantastic idea”, she opined that asking a person to do the driving simulator test right away would not be fair:

“*I think if you were to put someone behind it [driving simulator] immediately, I could see how someone would think it wouldn’t be fair because I don’t think it would be. It’s not the norm for them by any means being on a road versus on a computer screen … If this was the first time, I could see him [her husband] saying no it’s not going to be fair, unless there was that practice run.*”(P73)

#### 3.8.3. Good but Not for Everyone

Unlike the six participants who generally believed that the use of the driving simulator would have been beneficial in the case of their family members, two of the participants stated that the driving simulator test would not have worked for their family member with cognitive decline. Participant 3 was adamant that the driving simulator would not have been useful in the case of her husband. While she supported the idea of using the driving simulator, in principle, and called it a “terrific idea for people who want to do it or people that can handle it” and even stated that it should be used for everyone who is turning 70 and renewing his/her license, she believed that the use of the driving simulator would not have been useful in cases such as her husband’s because it was abundantly clear that her husband should not be driving and also because the use of the driving simulator would have made him angry and frustrated and increased the emotional burden on her and other family members.

“*No, no. He couldn’t drive. He drove into the garage once (laughs). He drove right into the back of our garage and smashed in. What [the doctor] did with the testing on the paper was more than enough…. No, I don’t feel that it would be useful for my husband. I think for a lot of people it would be great but not for my husband. His driving was so bad then, so it was too late.*”(P3)

Similarly, participant 87 voiced skepticism that the driving simulator would have been useful in the case of aging adults who struggle with technical tasks:

“*I’m always reluctant when we use computers to assess people … This is almost like a video game but not quite, so it’s just one of those things that I’m not so sure it would be good for her because she had problems assimilating technical stuff at her job …. Someone would have to prove to me that that test is an accurate representation of what is happening out there. Maybe in some conditions.*”(P87)

#### 3.8.4. Critique of the Current Testing System

While we did not explicitly ask that, half of the caregiver participants voiced criticism of the current testing system regarding aging adults. Specifically, they believed that the system is not rigorous enough: many aging adults are referred too late for assessments, and this is a problem because many continue to have valid driving licenses even though they pose safety hazard for themselves and others on the road. All of the participants in this study stated that they had misgivings related to PLWCD’s driving and/or getting lost while driving:

“*I’ve witnessed on quite a few occasions, not just with my mom, but with another elderly person that it just boggles my mind as to how it is that they were on the road because they were such a road hazard so safety for themselves and safety for others. The system seems quite broken when it comes to testing people, testing the elderly or testing anyone that is exhibiting any kind of cognitive or physical disabilities that put into question their ability to drive. I don’t think we do that very well.*”(P15)

### 3.9. Driving Simulator as Beneficial

All of the participants, regardless of whether or not they believed that the driving simulator would have been useful in the particular case of their family member living with cognitive decline, mentioned various benefits to the use of the driving simulator to test the driving skills of PLWCD. We identified three main sub-themes regarding the benefits of using the driving simulator: good indicator of driving skills, better than the alternative and makes it easier to understand driving license suspension.

#### 3.9.1. Good Indicator of Driving Skills

A number of respondents reported that the driving simulator is a good indicator of driving skills, with one of the participants commenting on how pilots are trained on simulators, and that we put our trust in pilots based on the fact that they passed the simulator tests:

“*So, if it’s presenting obstacles in the path that she has to react to and she’s not able to then to me that’s a good sign that she should not be on the road.*”(P15)

“*… you get a pretty good impression I guess of how the person is driving. It may not be 100% of the things you have to look for, but it’s damn close. It checks off a lot of boxes.*”(P49)

One participant though stated that it is important that the driving simulator assessment takes into account hand positioning and shoulder and mirror checks.

#### 3.9.2. Better Than the Alternative

Caregivers generally believed that the driving simulator is better than the alternative, such as written tests or road test. Caregivers were also in favor of the driving simulator because it offers a safer alternative to the road test. All caregivers in this study had expressed concerns related to the driving of the PLWCD even prior to the suspension. One of the participants noted that with the increase in the number of persons living with cognitive decline, it is important to have a good and safe option to assess their driving skills:

“*Absolutely. Absolutely. We went for written tests, but I don’t think that the written tests accurately test an individual’s driving capability which is what was in question.*”(P15)

“*… the numbers for dementia are going up so something would need to happen and better in advance in a safe setting than on the road.*”(P68)

#### 3.9.3. Makes it Easier to Understand the Suspension of Driving License

Several caregivers believed that the driving simulator would make it easier for the PLWCD to understand why their driving license was suspended. As discussed previously, caregivers reported that PLWCDs were baffled by the driving suspension, and none agreed with the assessment that they are no longer fit to drive. In this context, it is interesting to note that caregivers believed that driving simulator would make it easier to explain why the PLWCD failed the test. This could have the added benefit of reducing the pressure on physicians who are sometimes blamed by the PLWCDs for the loss of their driving license:

“*But I think it’s a good idea for people for two reasons. It could help them understand why their license was taken away from them. If they’re on a simulator and they’re leaving the lanes and if they can recognize the mistakes they’re making while doing this, it might ease them a bit the fact that they don’t drive anymore …*”(P3)

“*… it’s not subjective by the doctor, and I think it would remove from the doctor the bad responsibility to tell them you’re losing your driving license like I’m writing you to the MOT. Then people won’t maybe get mad at the [doctor] but maybe more mad at themselves that they failed the test …*”(P22)

### 3.10. Potential Drawbacks to Using the Driving Simulator

#### 3.10.1. Difficulty with Machine

Caregivers generally identified PLWCDs’ potential difficulty with the machine as one of the main drawbacks to the use of the driving simulators (one exception was in the case of PLWCD who was very familiar with simulators). Most of the participants noted that their family member with cognitive decline struggles with computers, smart phones or tablets, and that being in front of a simulator is unusual:

“*For somebody who is elderly, it is a very foreign thing to see a driving simulation like dealing with computers and so on. My mom doesn’t do that well.*”(P15)

“*I mean unless you sit down and all you have to do is gas, break and stir but if you have to turn it on or pause it or anything technical, she can’t do now like use her iPad or laptop or her cellphone.*”(P49)

#### 3.10.2. Practice Session

All of the participants were strongly in favor of a practice session prior to the test as it would give the PLWCD the opportunity to familiarize himself/herself with the simulator, get accustomed to it, become more comfortable, and more prepared for the actual test:

“*I think there would be for my mom, given her cognitive abilities, there would’ve been a difficulty to getting accustomed to the driving simulator. A practice session would definitely be helpful in her case.*”(P15)

“*He’s never done that before. He’s not computer literate …You and I could sit down and do it easily but people with dementia can’t. So, they need extra time for sure just to familiarize themselves with how this whole program worked to get their brain functioning on what it is they’re doing and that takes time.*”(P3)

#### 3.10.3. Doing the Driving Simulator Test Twice

Unlike the case for the practice session, the participants were roughly divided regarding having the PLWCD do the driving simulator test twice. Some participants firmly believed that there should be one test only for several reasons. Testing a second time could be a waste of time; it could lead to increased anxiety as PLWCD waits for the second appointment, and/or it might give PLWCD false hope, with the PLWCD wanting to take the test again and again:

“*I don’t know that I’d see the purpose of a second one … I think at this point I would start to question just how many times you would have to go in and do this … that she could continue to take tests until she got it right. So if you open up that door it leaves them potentially a false hope …*”(P15)

Other participants believed that having two tests is beneficial because it would allow participants to become more familiar with the simulator allows them to test better, and guard against the possibility that the result based on one test is a random event:

“*I think doing repeated testing would be better. With repeating it, it’s familiar with people. It lets them have that comfort level.*”(P73)

“*… the only way that this would work is if the doctor does not make a decision based on one test. If he or she indicated, we’re going to bring you back for another test just to compare the results … If you say this particular simulation test requires two visits to compare…*”(P87)

#### 3.10.4. Testing Anxiety

Each of the participants stated that the PLWCD would experience stress and/or anxiety with the driving simulator test; however, such anxiety is not related to the simulator per se, rather it is because of the issue of testing itself, and the fact that one’s driving license is at stake. PLWCDs would feel anxious because they could potentially lose their driving privileges, and consequently their independence:

“*My mom suffers from anxiety, well, as we all do, but it seems more acute in her case. Testing is a difficult thing for her…*”(P15)

“*I think she’d be stressed in a driving test whether it was a simulator or whether it was in real life. So, I think the stress would come from thinking about the appointment, worrying about the appointment, wondering if you know she’s gonna keep her license or lose her license and that would stress her out probably more than the actual test would.*”(P54)

#### 3.10.5. Crash in Simulator Stress

Participants generally believed a crash in the simulator would be upsetting to the PLWCD even though they believed that the PLWCD would understand that this is a simulation not a real-life accident. Some of the participants pointed out that a crash in the simulator could act as a reality check regarding the PLWCD’s driving.

“*I would rather have him do it on a simulator than on a road (laughs). The same thing can happen on the road as well. So, I still think a driving simulator would be more beneficial … Even though it’s just in the simulator, it would still bother him. He would consider that a failure …*”(P73)

“*… she would find it difficult, but I still believe that it should be there. She would not see this as omg I’m hitting somebody but instead, she’d come up with I’d never do that on the road. This is just a simulation this is not accurate. She would definitely recognize it as a simulation.*”(P15)

*”… I think that would be fairly terrifying for the driver. That would be awful … I don’t think now that my mother would think it’s real. If she had the test 2 years ago, she would probably understand by hitting the motorcyclist [in the simulator] that she’s not gonna get her license back …*”(P49)

“*… It could be traumatic but then it could turn into okay I won’t be the cause, or this affirms that I should not be driving, or I do not wanna cause harm to anyone or it might be a reality check. I think it goes without saying there shouldn’t be children in the simulation ....*”(P68)

#### 3.10.6. Additional Appointment

Most participants stated that an additional appointment to do the driving test simulator would not pose a burden. They pointed out that they have been going to numerous medical appointments, so an additional one is not a big deal. Additionally, participants seem keen to make use of additional appointments or new testing if it helps the PLWCD understand and deal better with the driving suspension:

“*I mean we were going through so many appointments then that one more wouldn’t have broken the camel’s back, and I think it might’ve helped her accept the situation a little bit better … I don’t think it would’ve been an issue.*”(P15)

Two participants were sort of the exception, and they stated that an additional appointment could be burdensome depending on the logistics, such as the location of the testing and difficulty finding parking:

“*… that could be a disadvantage. Especially the location like the physical aspects of it. It’s very hard to find parking down there.*”(P68)

### 3.11. PLWCDs’ Perceived Reactions to Simulator

The participants opined regarding how they believe the PLWCD would react to the simulator test results. This was informed by participants’ views regarding the simulator and their experiences with the PLWCD’s reactions regarding the driving suspension. As discussed previously, PLWCD were generally baffled by the driving suspension, and none agreed with the assessment that they are no longer fit to drive based on the written tests that they had done. We identified the following sub-themes regarding PLWCDs’ perceived reactions: failing would be upsetting; wanting to test again and acceptance of the results uncertain.

#### 3.11.1. Failing the Simulator: Upsetting

A number of participants believed that the PLWCD would be upset if he/she failed the simulator test. This is because PLWCD would lose their driving license and their independence, and they generally would undergo dramatic changes regarding their routines and way of living:

“*I think she’d still struggle with it in the sense that had she failed the simulation.*”(P15)

“*I think she still would’ve been angry … She’d still be really angry that her license was taken away.*”(P49)

#### 3.11.2. Wanting to Test Again

Many of the participants stated that the PLWCD would ask to be tested again, if he/she failed the simulator test. This is consistent with participants’ reports regarding the reactions of the PLWCD when he/she failed the written test:

“*I could see him possibly wanting to do another test. He would think automatically that he would be able to get his license back.*” (P3)

#### 3.11.3. PLWCD’s Acceptance of Results Uncertain

Participants were divided regarding whether the PLWCD would accept the results of the driving simulator. Some believed that the PLWCD would have a better reaction, would help PLWCD understand the suspension and, hence, he/she would accept it better than in the case of the written test as the simulator would give the PLWCD an idea of his/her safety on the road and have quite an impact on him/her even though he/she would grief the loss of the driving license:

“*I think for her it would’ve been a test with irrefutable results that would’ve said you know okay here’s the written test that you’ve had to go through and here’s a simulated driving situation and if you’ve not passed both…I think for her it would’ve helped her accept the situation ….*”(P15)

“*I think she would take it as gospel but still be annoyed that she can’t drive.*”(P49)

However, some of the other participants did not believe that the simulator would make a difference in the PLWCD’s reactions, and that he/she would come up with different excuses to discredit the testing and discount the results because in the words of one participant “nobody wants to admit that they can no longer drive, no one”:

“*I think it depends on the person’s personality. Somebody like my mom would probably say well that wasn’t real so how can you base this whether I lose my license from the fact that it’s not real.*”(P54)

“*If he wasn’t successful, he would’ve found an argument to be made. It’s just the way it is … He would’ve just found another way. He gets frustrated … it just wouldn’t have gone well.*”(P68)

## 4. Discussion

This study examined the experiences of family caregivers of persons living with cognitive decline (PLWCD) regarding driving cessation and their opinions about the use of driving simulators in driving assessment. Consistent with previous studies that have shown that caregivers are more likely to be women than men, the participants of this study were overwhelmingly women, and they were slightly more likely to be spouses than children. Their respective family members living with cognitive decline included equal numbers of men and women. This study shows that driving cessation is a significant disrupter in the lives of PLWCD and their caregivers. For the families in this study, driving cessation was an unplanned event, and the PLWCD and their caregivers generally experienced it as a crisis that led to PLWCDs’ feelings of anger, difficulty understanding the rationale for the suspension, and holding on to the belief that the suspension is temporary. Based on the caregivers, the driving cessation of the family member with cognitive impairment was associated with his/her feelings of loss of independence and loss of sense of control, isolation, coping struggles, and deterioration in the relationship between the PLWCD and the physician. The negative effects of the driving cessation on the PLWCD were exacerbated by the often sudden nature of the driving cessation and increased dependency on others due to limited or lack of alternative transportation options. Feelings of anger and deterioration in the relationship between the PLWCD and the physician were generally stronger among men than women.

All caregiver participants in this study had concerns related to PLWCD’s driving and/or getting lost prior to the driving cessation, and while most caregiver participants expressed relief that the PLWCD would no longer be driving, some caregivers experienced shock and stress due to the sudden nature of driving cessation and PLWCD’s strong negative reactions. Regardless of the family caregivers’ initial reactions, driving cessation generally had a significant negative impact on them due to PLWCD’s feelings of anger and blame toward the caregiver, increased tension and conflict in their relationship with the PLWCD, driving burden and exhaustion, difficulty in balancing caregiving duties with other responsibilities including work and caring for other family members, and having to navigate a strained relationship between the PLWCD and the doctor. Caregivers, particularly women caregivers, tended to bear the brunt regarding PLWCD’s driving cessation, whether in terms of the PLWCD’s anger and/or driving burden.

As for the driving simulator aspect of the study, caregivers were generally supportive of its implementation in driving assessment. However, support was nuanced in some cases. Some caregivers stressed the importance of a practice session in advance of the driving simulator test as well as transparency with the PLWCD and the caregiver regarding the impact of failing the simulator test on driving. Other caregivers noted that the simulator might not be a good solution for everyone, especially for those with advanced cognitive decline. Regardless of their position vis-a-vis the driving simulator, all caregivers believed that driving simulators had a number of benefits, including being a good indicator of driving skills, and a good alternative to memory tests and on-road tests. In addition, they felt that the results of a driving simulation could help explain the rationale of the suspension to the PLWCD, which could act as a buffer against increased strain on the relationship between the PLWCD and doctor.

Caregivers, however, recognized that the driving simulator has potential drawbacks, mainly, PLWCD’s difficulty with using the machine, anxiety caused by additional testing, the possibility of stress following a crash in the simulation, and the logistics regarding attending another appointment. Similarly, Matas and colleagues found that older adults may have difficulty using the driving simulator if they are unfamiliar with the technology [20]. Caregivers in this study were strongly in favor of a practice session just prior to the driving simulator test to give the PLWCD the opportunity to get familiar with the technology. However, caregivers were divided on whether the PLWCD should be offered the opportunity to take the driving simulator test again at a different appointment. While some caregivers believed that this could lead PLWCD to test better as he/she becomes more comfortable with the machine, others pointed out that the waiting time between appointments could be stressful for both the PLWCD and the caregiver. In addition, a second test could give the PLWCD false hope that they could keep on re-taking the driving simulator test until they pass. Stress due to crashing in the simulator is a possibility. In addition, caregivers generally expected the PLWCD to experience anxiety when doing the driving simulator test; however, such anxiety is due to the driving simulator per se; rather it is a common feature of any testing related to driving given its immense implications.

In terms of PLWCDs’ perceived reactions to the simulator test, many expressed the belief that the PLWCD would still be upset if he/she failed the test because of its implications on their lifestyle and independence, and some would ask to be tested again. While the use of the driving simulator could make it easier to explain the rationale of the suspension compared to cognitive tests, not all caregivers believed that the PLWCD would accept the results of the driving simulator test.

Driving gives people independence and autonomy, allows them to manage their schedules and needs and socialize and partake in activities that they enjoy [21]. As such, it is not surprising that driving cessation is associated with negative emotions [22]. Driving cessation leads to loss of independence, loss of control, increased dependency on others, and it possibly evokes fear and uncertainty of the future. This is particularly difficult for persons living in dementia who have been driving for decades. Our results are consistent with previous studies which reported feelings of anger and unjust treatment, loss of independence and sense of self and social isolation [9,10,11,23,24]. In our study, there seems to be gender differences in terms of reactions to driving cessation, with men generally expressing their reactions strongly and more negatively than women. Similarly, Adler and Rottunda reported that men found driving cessation more difficult than women [8]. Driving could be seen as a marker of masculinity particularly among older men, and their feelings could be exacerbated by the fact that it often falls on their caregivers, often women, to ensure that they are adhering to the driving suspension.

Adler reported that caregivers believed that physicians should take a leading role in addressing the issue of driving cessation with aging adults given their authority and expertise [6]. However, we generally found that driving cessation led to deterioration in the relationship between the physician and PLWCD, with the latter angry at and often refusing to see the physician again. Scott and colleagues found that PLWCD experienced driving cessation as a crisis, with general practitioners in Australia noting that PLWCDs who were told to stop driving often chose to transfer to other practitioners [25]. This is in line with an earlier study that found that the more a physician assumes a dominant role in the decision-making, the more likely it is for patients to blame the physician for a negative outcome [26]. PLWCDs’ anger at the doctor is likely due to the implications of the loss of driving privileges on their lives as well as feeling a sense of betrayal. Aging adults with dementia as well as some caregivers might not be aware that physicians and other healthcare providers in Ontario, Canada are legally obligated to report medical conditions that impair driving to the ministry of transportation [10].

As reported by Byszewski and colleagues, caregivers generally felt relief that the PLWCD no longer drives given the safety risks to them and others on the road [10]. However, the relationship between PLWCD and caregivers also comes under increased strain following driving cessation as the PLWCD often blames the caregiver who has to navigate the conflict in the relationship with the PLWCD along with an overwhelming increase in responsibilities to meet the needs of the PLWCDs. This is consistent with the literature showing that problems can develop between the PLWCD and their caregiver especially if it seems as though the caregiver helped trigger the issue of driving cessation [24]. Caregivers often feel they have to make sure that the PLWCD is no longer driving, and often they sell or move the car elsewhere. Connell and colleagues found that parents viewed adult children’s active engagement to stop them from driving as disrespectful, which some adult children found difficult to deal with [27]. Previous research has shown that elements, such as time off from work and providing transportation, contribute to caregiver burden [28]. There was also an influence on the caregivers’ ability on managing their own lives and responsibilities while also maintaining their caregiving duties. Previous studies have shown negative effects on caregivers [29], but this study showed that caregivers’ mental health was often directly impacted, often leading to worsening symptoms of depression and stress and even requiring the taking of medications. It is possible that the reactions in this study were more severe because most of this occurred during an already difficult time for PLWCD and caregivers, namely the COVID pandemic.

Given the negative impact of sudden driving cessation on the PLWCDs, caregivers and the relationship between the PLWCD and the physician, it is important that more attention is given to advanced planning and having aging adults and their families prepare for the possibility of driving cessation [21]. While all caregivers in the study were concerned about PLWCD’s driving, none reported discussions about advanced planning with the PLWCD regarding driving cessation. Scott and colleagues identified preparation and detailed planning, including alternative transportation options, as key to ease the PLWCD into driving cessation and normalizing the process [25]. Adler and Rottunda, similarly, found that few Americans make plans regarding driving cessation [8]. Betz and colleagues reported that clinicians do not initiate these discussions unless concerns were raised by family members or after notable changes in aging adults’ health. The PLWCD’s perceived negative reaction is seen as a barrier to initiating such discussions [11].

Having early and frequent conversations with aging adults regarding preparing for the possibility of driving cessation could ease the aging adult to stop driving as getting the PLWCD on board with driving cessation over a period of time is a much better alternative than a sudden and forced suspension. Adler reported that drivers with dementia who had numerous conversations with their family members about driving cessation tended to accept the decision [6]. Similarly, Byszewski and colleagues recommended “gradual disclosure and follow-up” [10] (p. 159) and Bryanton and colleagues reported that the process of driving cessation could have been improved if it were a more gradual one [30]. Healthcare providers could also benefit from additional training regarding approaching the issue of driving cessation with aging adults [11].

It is important that PLWCDs and caregivers believe that driving assessments are helpful and valid. Based on this study, PLWCDs and some caregivers were generally not convinced that the usual cognitive tests accurately reflect driving skills. This is not surprising given that the link between cognitive tests and driving assessments is not necessarily obvious. Driving simulators, along with cognitive tests, could instill more confidence and acceptance in the results. Driving simulators can test the drivers’ anticipation of hazards by subjecting them to different obstacles [31]. For obvious reasons, this is not possible to do this during an on-road test. In addition to its safety, its standardization of testing was noted by caregivers. Through this system, participants can be tested under the same conditions regardless of where they are. This is important given that the conditions experienced during an on-road test can vary for several reasons such as weather and traffic. Based on the caregivers, the simulator would also provide the PLWCD with the opportunity to better understand the reasoning behind the suspension of their driving license. A driving simulator test could allow the PLWCD to physically see the relation between the driving abilities they’re exhibiting during the evaluation and the physician’s decision behind reporting them to the driving authority. However, it is important that PLWCDs are given a practice session to allow them to become more familiar with the technology, and that physicians are aware that PLWCDs experience anxiety related to testing, and as such, it is important to ensure that the testing environment reduces PLWCD’s stress as much as possible.

While the participants provided rich data about their and the PLWCD’s experiences regarding driving cessation and the use of driving simulators to assess the driving fitness of PLWCD, one of the main limitations of the study is that it is based on a relatively small sample size. As mentioned previously, the study was based on the records kept by two physicians at Bruyère Memory Program in Ottawa. The list of potential participants consisted of the family member who had attended the appointment when driving assessment was discussed of every aging adult with cognitive decline who was reported to the MOT between 6 and 24 months before the time of the study and who had generic consent form to participate in future research studies in his/her file. While we contacted all potential participants based on the above criteria (32 in total), 8 (or 25%) provided informed consent and were successfully interviewed. The other potential participants either informed us that they could not participate because of their busy schedules or did not contact us back. Future studies would benefit from having a larger sample size, particularly to probe in more detail the gender differences in terms of PLWCD’s reactions reported in this study.

Subsequent studies should use the results obtained in this and previous studies as a baseline when expanding on this research topic. Furthermore, the perspectives of the persons living with dementia are another area of study where little research has been done. Although the progressiveness of the disease may impede the involvement of certain patients in the study, the nature of the project would only require their engagement at one point in time. Depending on the stage of their disease, PLWCD could still be recruited to offer their thoughts on the use of this technology. This would not only offer a new point of view, but it would also allow us to gain insight into how the specific group of people who would be using the technology view its potential implementation. Therefore, recruiting the PLWCD to get a firsthand account of their experiences and thoughts could arguably expand on what is known so far regarding the caregivers’ attitudes towards using driving simulators to assess driving fitness.

## 5. Conclusions

This study has two parts. The first confirmed the very significant impact on persons living with cognitive impairment and their caregivers when discussions around driving cessation happen, especially when this was unexpected. The reactions of caregivers in this study were in line with previous studies but seemed more severe. The second part explored family caregivers’ opinions about driving simulators. Overall, they felt that simulators would be a good addition to driving assessments for most persons living with cognitive decline, but they felt that a practice session would be important and that the care team should be prepared to deal with negative reactions if the PLWCD fails the test. The authors recommend that clinicians dealing with driving retirement continue to consider the very significant impact it has on persons living with cognitive decline and their families. As researchers continue to use driving simulators and other technology, they are encouraged to give older adults living with cognitive decline time to adapt to the technology.

## Figures and Tables

**Table 1 geriatrics-07-00126-t001:** Demographic characteristics of the participants in the study.

Participant ID	Gender of Caregiver	Age of Caregiver	Relationship to PLWCD	Gender of PLWCD
15	Woman	55–59	Daughter	Woman
22	Woman	60–64	Wife	Man
3	Woman	70–74	Wife	Man
49	Woman	60–64	Daughter	Woman
54	Man	55–59	Son	Woman
68	Woman	55–59	Wife	Man
73	Woman	60–64	Wife	Man
87	Man	70–74	Husband	Woman

**Table 2 geriatrics-07-00126-t002:** Summary of qualitative data analyses regarding driving cessation.

Topics	Themes	Sub-Themes
Reactions to PLWCD’s driving suspension:	Reactions of PLWCD	Negative emotions
		Difficulty comprehending rationale
		Thinking it is temporary
	Reactions of caregivers	Relief
		Shock and stress
Impact of PLWCD’s driving suspension:	Impact on PLWCD	Loss of independence and control
		Isolation
		Coping struggles
	Impact on PLWCD-doctor relationship	Anger at doctor
		Refusal to see doctor again
	Impact on PLWCD-caregiver relationship	Blame and anger
		Tension and stress
	Impact on caregivers	Driving burden and exhaustion
		Life-caregiving imbalance

**Table 3 geriatrics-07-00126-t003:** Summary of qualitative data analyses regarding caregivers’ opinions about the use of driving simulators to assess driving skills.

Themes	Sub-Themes
Usefulness of driving simulator	Unqualified support
	Conditional support
	Good but not for everyone
Driving simulator as beneficial	Good indicator of driving skills
	Better than alternative Makes it easier to explain suspension
Driving simulator’s potential drawbacks	Difficulty with machine
	Testing anxiety
	Crash stress
PLWCD’s perceived reactions to simulator	Failing simulator upsetting
	Wanting to test again Acceptance of results uncertain

## Data Availability

Not applicable.

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
