# Peer review of "Driving Cessation: What Are Family Members’ Experiences and What Do They Think about Driving Simulators?"

_geriatrics, 2022, doi:10.3390/geriatrics7060126_

Round 1

Reviewer 1 Report

Manuscript: geriatrics-1975677
Driving cessation: What are family members’ experiences and what do they think about driving simulators?

1. Originality:
This is an interesting manuscript that can contribute to research in public health, population health, and social science. Despite the importance of the topic, there are some major conceptual and methodological issues with writing clarity that need to be clarified and addressed in this manuscript. A major revision of this manuscript is required to bring the manuscript up to qualifying publication.

2. Introduction:
- In the introduction, the authors unclearly expressed the significance of the study and failed to address the theoretical underpinning of this research. Also, it is necessary to clarify the concept of driving cessation, especially in people with cognitive decline. The authors should strongly address the gap of this study, particularly in the context in which this study was conducted. What is exactly new in this study?

- Several sentences need to be cited, especially in the 1st paragraph (Lines 48-54) and the 4th paragraph (Lines 75-90).

3. Methodology:
There are some methodological issues in this manuscript that necessary to be clarified and addressed.
- Some of the main open-ended questions in the in-depth interviews should be provided.
- Please provide the details of the sample size and sampling procedure. These steps are necessary to be described in more detail.

- Trustworthiness/Rigor (credibility, transferability, dependability, and conformability) should be mentioned as a subheading in this section.
- The ethical issue there is little attention given to the ethical issues. Which ethical practices that the researchers have done in this research? Please provide the details of these issues. Also,

- The Institutional Review Board Statement needs to be mentioned regarding the journal guidelines at the end of this manuscript (before the references section).

4. Results and Discussion:
- Please kindly check the typing error and the in-text citation formatting. Also, check the references style regarding the journal guidelines in “MDPI Chicago Style.”
- The discussion section is promising, but the author seems to have overstressed what was described in the results section without convincing the reader about the study's robustness.

- The conclusion needs to be rewritten and substantiated by the data. In conclusion, please mention all key findings you can conclude from this research.
- Please add more “Recommendations”. The authors may think about the stakeholders, persons, or agencies that the authors need to communicate the findings from this research with.

Reviewer 2 Report

Manuscript ID: geriatrics-1975677Type of manuscript: ArticleTitle: Driving cessation: What are family members’ experiences and what do they think about driving simulators?

Thank you for the opportunity to review this interesting study manuscript. This manuscript explores an important topic that has relevance to readers of the Ageing and Driving special edition, i.e., the impact of driving cessation on persons living with cognitive decline. Specifically, this qualitative study reports the perspectives of a sample of 8 family caregivers of persons with cognitive decline, who participated in interview about driving assessment, and further explores their opinions about use of driving simulators in assessing fitness to drive. The manuscript is clearly written and well-structured, with occasional typographical errors only. The qualitative methods are appropriate to answer the research questions. The conclusions drawn are consistent with the data and are supported by the included citations.  

Below, please find some suggested minor corrections and recommendations.

The statement at line 92 reads “not much is known about what “older drivers” and their family members think about this method of assessment.”, it is suggested that the statement be rephrased to represent the study sample and rationale, i.e. the focus on older drivers with cognitive decline. 

Line 94, Could the authors elaborate on the first research question [“1) get further understanding of the events surrounding the appointment when driving cessation was discussed”], specifically to clarify what is meant by the “appointment” and what “events” refer to. From further reading, it will be apparent to readers that it refers to events or discussions between family caregivers and persons with cognitive decline around the appointment time that driving risk was discussed, however the reader would benefit from knowing this at this early point in the manuscript, i.e. line 94.

Line 190 states that “There were four male and four female older adult drivers with cognitive decline. Seven were reported to the Ministry of Transportation, and subsequently stopped driving, and one did an on-road test that they passed.” Although participant IDs are used throughout the quoted segments and there is good distribution of these example quotes across the sample, it would be beneficial to have a link between these participant IDs and certain demographic information in a table, such as knowing the driving status (as per the above statement at line 190), and/or the caregiver relationship to PLWCD, such as knowing if they are partner or child. If revealing these types of details poses an issue to anonymity, might the authors then consider providing an overall statement about whether there were any differences [in caregiver reports] regarding the driver who passed the driving assessment, e.g. their emotional reactions, anger, grief, relationship impact?

Lines 210 and 213, use both acronyms PLWCD and PLWD. It is unclear whether this is intentional, nevertheless it is recommended that one or other is used, and for consistency throughout the paper.

Table 1, also refers to PLWD, and not PLWCD as per the body of the manuscript.

Methods: a recommendation is to include the list of interview questions, either in a table or as online materials for the benefit of readers.  

Lines 102-103 include the ethics statement, "Research Ethics approval was obtained from three research ethics boards: Bruyère Continuing Care, University of Ottawa and Carleton University", the available ethics approval ID/numbers might be included also.

Discussion: There are no limitations discussed, which might be an oversight. For example, sample size, although a qualitative study, it may still be considered small; further, only one participant had passed the on-road test, while all 7 others had failed, might the results have varied if there were an even representation of those that passed and did not pass the driving assessment?

Thank you.

Round 2

Reviewer 1 Report

Congratulations to the Authors as they properly addressed all of my previous comments, and I believe that their article is now ready for publication.